# Multi-Stage Learning Framework Using Convolutional Neural Network and Decision Tree-Based Classification for Detection of DDoS Pandemic Attacks in SDN-Based SCADA Systems

**DOI:** 10.3390/s24031040

**Published:** 2024-02-05

**Authors:** Onur Polat, Muammer Türkoğlu, Hüseyin Polat, Saadin Oyucu, Hüseyin Üzen, Fahri Yardımcı, Ahmet Aksöz

**Affiliations:** 1Department of Computer Engineering, Bingöl University, Bingöl 12000, Turkey; huzen@bingol.edu.tr; 2Department of Software Engineering, Samsun University, Samsun 55000, Turkey; muammer.turkoglu@samsun.edu.tr; 3Department of Computer Engineering, Faculty of Technology, Gazi University, Ankara 06500, Turkey; polath@gazi.edu.tr; 4Department of Computer Engineering, Adiyaman University, Adiyaman 02040, Turkey; saadinoyucu@adiyaman.edu.tr; 5Independent Researcher, Ankara 06500, Turkey; 6MOBILERS, Sivas Cumhuriyet University, Sivas 58580, Turkey; aaksoz@cumhuriyet.edu.tr

**Keywords:** SCADA, SDN, cyber pandemic, DDoS attacks, CNN, machine learning, critical infrastructures

## Abstract

Supervisory Control and Data Acquisition (SCADA) systems, which play a critical role in monitoring, managing, and controlling industrial processes, face flexibility, scalability, and management difficulties arising from traditional network structures. Software-defined networking (SDN) offers a new opportunity to overcome the challenges traditional SCADA networks face, based on the concept of separating the control and data plane. Although integrating the SDN architecture into SCADA systems offers many advantages, it cannot address security concerns against cyber-attacks such as a distributed denial of service (DDoS). The fact that SDN has centralized management and programmability features causes attackers to carry out attacks that specifically target the SDN controller and data plane. If DDoS attacks against the SDN-based SCADA network are not detected and precautions are not taken, they can cause chaos and have terrible consequences. By detecting a possible DDoS attack at an early stage, security measures that can reduce the impact of the attack can be taken immediately, and the likelihood of being a direct victim of the attack decreases. This study proposes a multi-stage learning model using a 1-dimensional convolutional neural network (1D-CNN) and decision tree-based classification to detect DDoS attacks in SDN-based SCADA systems effectively. A new dataset containing various attack scenarios on a specific experimental network topology was created to be used in the training and testing phases of this model. According to the experimental results of this study, the proposed model achieved a 97.8% accuracy rate in DDoS-attack detection. The proposed multi-stage learning model shows that high-performance results can be achieved in detecting DDoS attacks against SDN-based SCADA systems.

## 1. Introduction

SCADA is a hardware and software system that monitors and controls industrial processes in real-time from local or remote locations. It is used in various industries, including defense, transportation, energy, automotive, healthcare, and critical infrastructure. This system consists of four main components: a communication network, a Master Terminal Unit (MTU), a Human–Machine Interface (HMI), and Remote Terminal Units (RTUs) equipped with sensors/actuators. Traditional SCADA networks can be complex, making network management difficult. They are also inflexible when adding or replacing devices, and interoperability between different vendor devices is limited. Network operators must use vendor-specific policies to implement complex configurations [1].

Traditional SCADA systems face challenges in keeping up with rapidly changing information technologies and require significant costs and labor to maintain vendor-specific equipment and software upgrades. To modernize these systems and adapt to Industry 4.0 trends, Software-defined networking (SDN) offers a promising solution. By separating the control and data planes and delegating critical control functions to an SDN controller, SDN provides a software-based centralized network control mechanism that simplifies large-scale network management. SDN focuses on flow routing and increases network programmability, making it an efficient alternative to traditional network configurations. SDN uses open standards, allowing for the flexible configuration and maintenance of network elements. It performs better than traditional networks under changing traffic conditions, with lower packet latency and jitter [2].

SDN-based SCADA systems offer enhanced security compared to traditional SCADA systems. They provide benefits such as network topology management, centralized access control, and effective security policy enforcement [3]. However, the separation of control and data planes in SDN-based SCADA networks introduces new security concerns. The first concern is cyber-attacks targeting the SDN controller in the control plane, which can bring down the entire system. The second concern is attacks on the communication channel between the control and data planes, leading to packet loss and network unavailability [4]. The most dangerous attacks in SDN-based SCADA networks are DDoS attacks, which consume resources and target availability, causing performance degradation, data loss, and service outages. Even attackers with minimal technical skills can often successfully launch DDoS attacks. Developments in information technology, individuals’ retreat to their homes during the COVID-19 pandemic, and the increasing shift to remote working have made DDoS attacks more effective, complex, widespread, high volume, and vector-oriented than ever before [5]. In particular, it has become quite difficult to distinguish between a legitimate increase in internet traffic and a DDoS attack. Very large DDoS attacks occurred between July and September 2020, reaching nearly 1 Tbps [6]. The pandemic also saw the return of an old threat type, the Ransom Distributed Denial of Service (RDDoS) campaign. Attackers can threaten organizations with a DDoS attack unless a ransom is paid. These types of ransom-related DDoS attacks are becoming more common because they require less effort than installing malware on an organization’s IT infrastructure [7].

DDoS attacks have now become a global risk, affecting not only individuals and businesses but also communities, cities, and countries, turning into a cyber pandemic. A DDoS threat consistently ranks high in the European Union Agency for Cybersecurity (ENISA) Threat Landscape due to its high impact potential [8].

SCADA networks are a prime target for cybercriminals seeking to disrupt critical infrastructure operations such as industrial, electrical, water treatment and distribution, natural gas and oil, transportation, and healthcare systems. If a DDoS attack is launched against an SDN-based SCADA network that does not have real-time and automated DDoS-attack-detection capabilities, it can cause massive chaos and have catastrophic consequences. DDoS attacks do not actually attempt to obtain sensitive information from a system. The goal is to slow or stop the use of the target system. Any disruption to SCADA networks due to cyber-attacks such as a DDoS can threaten human safety, cause a loss of productivity, economic loss, a loss of reputation, and even some environmental damage [9].

The time-sensitive nature of a service disruption to a SCADA system requires immediate action to restore the system as quickly as possible. Therefore, the real-time detection of DDoS attacks against SDN-based SCADA systems is critical. The primary challenge in detecting a DDoS attack is discerning the attack traffic from the regular traffic [9]. With advanced security solutions capable of real-time detection and analysis driven by artificial intelligence, SDN-based SCADA networks can be continuously monitored for suspicious and abnormal activity [10]. By detecting a possible DDoS attack at an earlier stage, security measures that can reduce the impact of the attack can be taken immediately, and the likelihood of being a direct victim of the attack will be reduced [11].

This article discusses a multi-stage learning model proposed for effectively detecting DDoS attacks on SDN-based SCADA systems. The proposed model integrates 1D convolutional neural network (1D-CNN) and decision tree methods. Based on a specific experimental network topology, a dataset containing regular and DDoS-attack traffic specific to SCADA systems was generated to be used in the training and testing phases of the model. According to the experimental results of this study, the proposed model achieved 97.8% accuracy in detecting DDoS attacks. This result demonstrates that the proposed multi-stage learning model can successfully identify DDoS attacks on SDN-based SCADA systems with a high success rate.

The proposed model can be an important step in detecting DDoS attacks with high accuracy, thus helping to make systems more secure.

### 1.1. Perspective and Goals

The integration of new technologies, such as SDN, into SCADA systems has revolutionized industrial environments. These advanced systems have become essential for the efficient and effective operation of critical infrastructure, such as power plants, water treatment facilities, and transportation networks. However, the threat of cyber-attacks, such as DDoS on SDN-based SCADA systems, cannot be ignored. If left undetected, these attacks can cause significant economic loss and extensive damage to the attacked system. Therefore, it is crucial to take appropriate measures to protect these systems from cyber threats. Developing an accurate and sensitive DDoS-attack-detection system based on artificial intelligence is a critical step in ensuring the effectiveness and security of industrial systems against cyber-attacks.

The current study proposes a hybrid model that combines deep learning and traditional machine learning approaches to detect DDoS attacks. Deep learning is powerful on large datasets, while traditional machine learning is effective in more specialized scenarios. By combining the advantages of both approaches, this hybrid model can help detect a broader range of attacks and provide a more robust solution from a security perspective.

The main contributions of this study are as follows:SDN technology was used in this study, one of the new intelligent technologies recommended to overcome the problems experienced in traditional SCADA systems, such as manageability, service quality, and optimization. In this regard, it raises awareness.The dataset used in this study was obtained by creating a specially designed simulated-based experimental SDN-based SCADA architecture. The obtained dataset contains features specific to SDN-based SCADA architecture.This study presents a multi-class deep learning and decision tree-based classification approach to identify DDoS attacks on SDN-based SCADA systems. The presented model aims to make identifying DDoS attacks more sensitive and reliable.This study focuses on detecting DDoS attacks, which is a significant challenge for the security of industrial control systems. This aims to contribute to the efforts of industrial enterprises to protect their critical infrastructure.This study presents experimental results of the proposed approach, confirming that this approach can effectively detect DDoS attacks. These results offer the possibility of use in real-world applications for security experts and network administrators.

### 1.2. Organization of This Study

This study is organized into six chapters, as shown in Figure 1. Section 2 examines similar studies in the literature on detecting cyber-attacks targeting SCADAs. Section 3 discusses the SDN-based SCADA system architecture, and the main targets and impacts of DDoS attacks on SCADA systems. Section 4 details the materials and methods used in this study, such as the dataset and the deep learning model architecture. The experimental study results are given in Section 5. Finally, Section 6 summarizes the results of this study.

## 2. Previous Studies on the Security of SCADA

After the rapid development of information and communication technologies, machine learning and deep learning techniques have become ubiquitous in identifying DDoS attacks targeting industrial systems. Below, we present a summary of some relevant studies.

With Industry 4.0, new-generation information technologies have been transformed into interconnected innovative Cyber–Physical Production Systems. Although this digitalization enables efficient, fast, and effective production, these systems, consisting of thousands of nodes, have become targets, especially against DDoS attacks. Saghezchi, F.B. et al. proposed eleven different unsupervised, semi-supervised, and supervised machine learning algorithms to detect cyber-attacks on Cyber–Physical Production Systems. According to the experimental results, it has been observed that supervised learning algorithms perform better than both unsupervised and semi-supervised algorithms in anomaly detection [10].

Artificial intelligence methods prevent attacks against resource-constrained IoT devices, especially those used in industrial systems. In their study, Wang, Z. et al. proposed a deep learning model to detect attacks on resource-constrained IoT devices used in industrial systems. The proposed model comprises four main components: feature selection, deep metric learning, a Knowledge Distillation model based on Triplet CNN, and a novel neural network training method named K-fold cross-training. This study aims to improve model performance, increase anomaly-detection speed in industrial systems, and reduce the complexity of the model [12].

To efficiently use the new generation of intelligent electricity-distribution networks shaped by the integration of communication systems, adequate measures must be taken against cyber-attacks. Diaba, S.Y. et al. introduced a hybrid deep learning model that utilizes CNN and Gated Recurrent Unit (GRU) algorithms to detect DDoS cyber-attacks against the communication infrastructure of the Smart Grid. This study employed the dataset CIC-IDS2017 from the Canadian Institute of Cybersecurity for classifying DDoS attacks. The hybrid model presented by the authors achieved a remarkable overall accuracy of 99.7% [13].

In their study, Wang, W. et al. proposed a stacked deep learning method to ensure the productivity and security of industrial systems against cyber-attacks. This study used machine learning algorithms such as Xtreme Gradient Boosting (XGBoost), Random Forest, Support Vector Machines, Linear Discriminate Analysis, and Conditional Inference Tree. The analysis utilized a dataset encompassing various forms of cyber-attacks on SCADA systems within laboratory-scale gas pipelines and water storage facilities. According to the experimental results obtained in this study, the XGBoost machine learning algorithm was found to be more successful in detecting attacks than other machine learning models and the proposed deep learning method [14].

Ferrag, M.A. et al. proposed a deep learning-based intrusion-detection system that utilizes CNN, deep neural networks (DNNs), and Recurrent Neural Networks (RNNs) to identify DDoS attacks on Smart Agriculture or Agricultural IoT applications. This study evaluated the performance of each model by analyzing the CIC-DDoS2019 and TON_IoT datasets, both of which comprise various types of DDoS attacks. The CNN-based model achieved the highest accuracy of 99.95% and 99.92% for both datasets [15].

Wang, W. et al. suggested a way to detect cyber-attacks on SCADA systems using a deep learning-based technique. To train classifier models and assess their testing performance, the research team leveraged datasets from two laboratory-sized SCADA systems: a power transmission system and a gas pipeline. According to the authors, the stacked deep learning model outperformed conventional machine learning algorithms in detecting cyber-attacks [16].

Mohammed, A.S. et al. have asserted that the use of vulnerable protocols such as MODBUS, DNP3, and OPC in critical systems like SCADA can heighten the risk of cyber-attacks. To address this issue, these researchers proposed a supervised machine learning mechanism that utilizes the XGBoost algorithm to detect DoS attacks that exploit the weaknesses of these protocols. The results of this experiment indicate that the suggested approach can accurately identify these attacks with 99% precision [17].

New technologies have been integrated into industrial and cyber–physical systems with the modernization of industrial environments. The diversity of devices and applications on the system has made these systems a target for attackers. Fernandez O. et al. proposed a deep autoencoder-based intrusion-detection system trained on network flow data that does not require prior knowledge of the network topology against these attacks on industrial and cyber–physical systems. The developed model showed a successful performance, especially against DDoS attacks. In addition, this model also provides the opportunity to detect anomalies of legitimate nodes found in the network after the attack [18].

Altaha and Hong introduced the Function Code Autoencoder IDS (FC-AE-IDS), an intrusion detection system that uses unsupervised deep learning for DNP3 systems, one of the most prevalent protocols in SCADA systems. Its primary objective is to prevent servers from being captured by attackers and escaping rule-based packet inspection. According to the authors, their proposed approach achieves over 95% detection accuracy for all attack scenarios [19].

Khan, F. et al. stated, in their study, that integrating new technologies, such as the IoT, into industrial systems brings new attack threats. In addition, this study emphasized that more than traditional security methods are needed in new-generation industrial systems equipped with new technologies. For this reason, a technique has been proposed to detect attacks on SCADA-based IoT systems using deep learning-based Pyramidal Recurrent Units (PRUs) and decision tree. The proposed approach was assessed using 15 datasets obtained from SCADA-based networks. The experiment results demonstrate that the proposed strategy surpasses traditional techniques and detection approaches based on machine learning [20].

In conclusion, previous studies on the security of SCADA unveil a nuanced panorama of contemporary endeavors aimed at fortifying industrial and cyber–physical systems against the escalating threat of DDoS attacks. The rapid evolution of information and communication technologies has propelled the integration of machine learning and deep learning techniques into the forefront of defensive strategies. Industry 4.0’s transformative impact is evident as new-generation information technologies metamorphose into interconnected Cyber–Physical Production Systems (CPPSs) [10], enabling efficient production but also exposing these systems to heightened vulnerabilities, particularly against DDoS attacks.

Noteworthy contributions from various studies in the literature elucidate innovative approaches, ranging from supervised learning algorithms for anomaly detection in CPPS to deep learning models addressing attacks on resource-constrained IoT devices, hybrid deep learning models achieving remarkable accuracy in detecting DDoS cyber-attacks, and the exploration of stacked deep learning methods in laboratory-scale SCADA systems. The integration of new technologies and novel defensive measures underscores the imperative of transcending traditional security methods in the face of evolving cyber threats. Collectively, previous studies on the security of SCADA encapsulate the current state of research, emphasizing the need for adaptive and innovative security measures to safeguard critical industrial infrastructures in an era dominated by technological advancements and persistent cyber threats.

## 3. SDN-Based SCADA Systems

Although technologies in communication and information, particularly cloud technology, big data, mobiles, and the IoT, have advanced significantly in the last ten years, the principles of traditional networks have remained unchanged. Typically, the traditional network structure adopts a hierarchical design with Ethernet switch layers. This design proved acceptable in networks where a client–server architecture was extensively deployed. However, static architectures have approached operational limits for data centers, campuses, industrial systems, and mobile networks requiring dynamic computing and storage [21]. The network industry has been forced to reassess traditional structures due to rising trends, including the proliferation of mobile devices, server virtualization, and the inception of cloud computing [22]. Owing to the increasing bandwidth, a high connection speed, and accessibility requirements, as well as dynamic management demands of modern information and communication technologies, conventional network structures are confronted with complexity and manageability issues. It is now difficult to meet current market needs with the traditional network structure [23].

The rapidly developing and changing variety of products and services necessitates simultaneous progress in network management. Security, performance management, and control are essential elements in network management technologies. In traditional networks, network management is quite tricky. Since many types of devices exist in conventional networks, such as routers, switches, firewalls, load balancers, and intrusion-detection systems, scaling and securing the network is challenging. In addition, the network has a heterogeneous structure, because different manufacturers provide the devices, applications, and services on the network. Requiring low-level, vendor-specific configurations to implement high-level network policies introduces an additional layer of complexity. Due to the increasing size, heterogeneity, and complexity of traditional networks, traditional approaches will need to be revised for configuring, optimizing, and troubleshooting networks shortly [24].

Despite numerous proposals aimed at simplifying the administration of traditional networks, various network-management issues persist due to the challenge of altering the fundamental structure.

Existing SCADA systems, which have become more innovative with Industry 4.0, have become unable to meet the increasing requirements regarding management, scalability, and performance. SCADA systems must be integrated with new technologies, such as SDN, which has a programmable, flexible, and dynamic architecture (Figure 2). Energy efficiency, environmental sustainability, and economic performance can be increased by eliminating the high complexity in industrial systems with new generation networks to be created with SDN-based SCADA systems [24].

## 4. Methodology and Experimental Setup

### 4.1. Creating the Dataset

The dataset comprises 89 features, containing data that are unique to the SDN-based SCADA network. In total, there are 2725 data samples, among which 720 correspond to normal-network-traffic data, while 2005 correspond to DDoS-attack data [25].

The experimental topology is illustrated in Figure 3 and is designed to capture data on both regular-network and DDoS-attack traffic within the SDN-based SCADA network. The dataset includes specific information related to the SDN architecture, such as messages exchanged between the controller and network devices, non-matching flows, flow modification messages from the controller, flow table entries, the CPU utilization of transmission equipment, and average latency for flow creation. The experimental topology was conducted on a principal computer system with 32 GB RAM and an Intel i7-1165g7 processor operating on the Ubuntu 20.04 LTS system. Network data were collected using sFlow-RT from the OVS switch during both attack and non-attack scenarios. The collected data were stored in an InfluxDB database using JavaScript codes with timestamps for further analysis.

The Telegraf software(v1) is installed on Host 4, collecting telemetric values like CPU and memory through the Modbus-TCP protocol. sFlow-RT on the main machine records this data in the InfluxDB database. DDoS attacks are simulated using the Hping3 packet generator tool, generating attack packets with randomly spoofed source IP addresses and utilizing UDP, TCP, and ICMP protocols. Host 2, identified as the attacker (IP: 10.0.0.2), uses Hping3, targeting Host 4 as the victim (IP: 10.0.0.10). The SDN-based SCADA topology incorporates the Modbus messaging protocol, with Host 3 (IP: 10.0.0.3) acting as the Modbus master and Host 4 as the Modbus slave in the created network.

The dataset was generated through a four-step scenario conducted within a 60-min timeframe for TCP, UDP, and ICMP packets. Each packet, with a size of 512 bytes, was initially sent as regular and then as malicious packets using UDP, TCP, and ICMP protocols. The attack simulation involved Hping3 sending over 2000 packets per second. Communication in the SDN-based SCADA network began with the Modbus master (Host 3) establishing a connection with the Modbus slave (Host 4). After receiving registered values from the slave node to the master node, the dataset was obtained by sequentially running the four scenarios.

*Scenario 1:* During Modbus communication between Host 3 and Host 4, the Host 2 user launches a TCP flood attack against the Host 4 user.

*Scenario 2:* During ongoing Modbus communication between Host 3 and Host 4, a UDP flood attack is initiated by the user of Host 2 against the user of Host 4.

*Scenario 3:* During ongoing Modbus communication between Host 3 and Host 4, a user from Host 2 initiated an ICMP flood attack against a user on Host 4.

*Scenario 4:* While Modbus communication continues between Host 3 and Host 4, ping packets are sent from the Host 1 user to the Host 4 user, ensuring regular network traffic.

### 4.2. Methodology

Choosing an appropriate model is a critical element in solving data science problems. The type of model to be used should be appropriate to the nature of the problem, the structure of the dataset, and the purpose of the solution. This study aims to detect DDoS attacks with higher accuracy in SDN-based SCADA systems with a model built using a multi-stage learning network (MS-LNet) framework. MS-LNet combines different learning strategies and model structures at each stage. The MS-LNet framework proposed in this study involves creating a new, more comprehensive hybrid model by integrating the 1D-CNN deep learning model and the decision tree machine learning model. The reason for choosing a new hybrid model created by combining two models, 1D-CNN and decision tree, using a multi-stage learning approach is that it effectively integrates the advantages of 1D-CNN and decision tree models. In this proposed model, the features extracted by 1D-CNN are fed as input to the decision tree model. While 1D-CNN is used to extract deeper and more complex features, decision tree is more effective in decision making by considering these features in a more interpretable form.

The dataset in DDoS-attack-detection applications is in a tabular data structure. This structure consists of rows and columns; each row typically corresponds to a sample, while each column represents a specific feature. Decision tree, which is easy to use and understand, has long been used to build models from tabular data and, in many cases, can still achieve more successful results than other machine learning and deep learning models. Although deep learning methods have shown tremendous success with images, audio, and text, traditional decision tree models still have great potential in terms of performance with tabular data [26].

The first experiments conducted in this study include analyzing the performance of the decision tree machine learning model and the 1D-CNN deep learning model and investigating whether these models are a good choice. In the training and testing phases of both models, a dataset containing regular and DDoS-attack traffic specific to SDN-based SCADA systems was used based on a specific experimental network topology. In the experimental studies, the dataset is divided into training and test sets of 80% and 20%, respectively.

In the first experiment, a decision tree model for DDoS attack classification was built using the dataset specifically designed for this study. In this study, the parameters of the decision tree are Gini: max depth, 12 and min sample leaf, 14. The confusion matrix obtained from this experimental study is shown in Figure 4.

As shown in Figure 4, the accuracy, recall, precision, F1-score, and specificity performance values of the decision tree model are 95.23%, 95.23%, 95.24%, 95.23%, and 98.36%, respectively.

In the second experiment, the performance analysis of DDoS detection was performed using a 1D-CNN model (see Figure 5, Stage 1). One-dimensional convolutional neural network is a highly effective model for feature-extraction and classification tasks on time series, text data, or other one-dimensional data structures. In the 1D-CNN architecture, classification is performed at the end of the convolutional layers by distinguishing patterns belonging to different classes, such as DDoS attacks or regular network traffic, through fully connected layers and a softmax layer. The training phase of the 1D-CNN, the batch size, and the epoch number are set to 128 and 300, respectively. Adam optimization is used to optimize the parameters of the network. The resulting confusion matrix is shown in Figure 6.

As shown in Figure 6, the accuracy, recall, precision, F1-score, and specificity performance values of the 1D-CNN model are 94.13%, 94.13%, 95.07%, 94.02%, and 97.96%, respectively.

Comparing the test results of the decision tree and 1D-CNN models, it can be seen that although the 1D-CNN model is effective, it achieves lower classification accuracy than the traditional decision tree model. These results are due to some weaknesses of the softmax and fully connected layers used in the 1D-CNN architecture. In the 1D-CNN model, fully connected layers do not take into account local structures of the input (e.g., sequential features in time series). Therefore, the 1D-CNN model may have difficulty fully capturing patterns in the data.

After these two experiments, we focused on a model structure that could further exceed the test performance results of both the decision tree and 1D-CNN models. We decided to create the 1D-CNN+decision tree model by integrating both models. The new hybrid model obtained by combining two models, such as 1D-CNN and decision tree using a multi-stage learning approach, has shown that this has the potential to provide a better generalization and better performance in experimental studies.

### 4.3. Creation of 1D-CNN + Decision Tree Model with MS-LNet Framework

The multi-stage learning network (MS-LNet) framework includes different stages where a new hybrid model is obtained by combining the 1D-CNN and decision tree models. The MS-LNet framework proposed in this study consists of 3 stages (Figure 5).

Stage 1: This is the stage where the 1D-CNN model is trained.Stage 2: While training the 1D-CNN model, the 1D-CNN and the decision tree models are combined. The feature extraction capabilities of the 1D-CNN model are combined with the decision tree model. At this stage, the weights of the 1D-CNN model are frozen, so only the decision tree model is trained.Stage 3: The 1D-CNN and decision tree models are combined, the weights of both models are frozen, and the resulting new model is tested.

As shown in Figure 5, in the first stage of the proposed MS-LNet model, a CNN architecture was trained. This CNN architecture includes three convolutional layers and two fully connected layers.

In the second stage, features were extracted using the fully connected layers of this trained CNN architecture. Then, these deep features were used to train the decision tree classifier. In the last step, the accuracy of the proposed model was evaluated using examples from the test dataset. In summary, CNN is used to extract important features with the ability to process input data, while decision tree is used to classify these features and determine attack types. Thus, the combination of the CNN and decision tree methods aims to detect DDoS attacks in SDN-based SCADA systems more effectively and accurately. In the following sections of this study, the stages of the proposed model are described in more detail.


**Stage 1: Proposed CNN model**


Since the development of deep learning, it has been actively applied in many fields. Unlike traditional machine learning models, deep learning architectures use a learning-based approach to extract appropriate features. This approach uses convolutional layers to extract features from input features. The convolutional layer is essentially a filtering process. In 1D-CNN, one or more filters are used. Each filter creates the feature map by performing a shift operation on the data. Specifically, these filters are used to emphasize or detect certain features. Filters are used to scan the data using a sliding window. Mathematically, the convolution process is as follows (Equation (1)):(1)$$\left(X*k\right)\left[i\right]=\sum_{j=1}^{k}X\left[i+j-1\right].k[j]$$

Here, *X* represents the input data, *k* represents the filter vector, and (*X*
∗
*k*)[*i*] represents the *i*th output element of the processing result. Activation functions are applied to the outputs obtained by convolution [27]. This helps the model learn nonlinear features of the data.

The resulting feature map is given to one or more fully connected layers. These layers are used for classification or regression operations. Each neuron can represent different features of the feature map. Finally, a softmax layer is added to the output layer of the model to produce classification results. The characteristic features of the CNN architecture used in the current study are shown in Table 1. In addition, an example of the developed CNN architecture is shown in Figure 5a.

As shown in Table 1 and Figure 5a, three convolution layers and three fully connected layers were used in the proposed 1D-CNN architecture. The ReLU activation function was used at the end of each convolutional and fully connected layer. The softmax classifier was used in the last layer of the network architecture.

In the first stage of the proposed MS-LNet model, the CNN architecture was trained with the dataset. The Binary Cross-Entropy loss function was used for training. The Binary Cross-Entropy loss function is defined as in Equation (2).
(2)$$L\left(y,\hat{y}\right)=-\frac{1}{N}\sum_{i=1}^{k}\left[y_{i}\cdot \log\left({\hat{y}}_{i}\right)+\left(1-y_{i}\right)\cdot log(1-{\hat{y}}_{i})\right]$$

Here, *L(y,*$\hat{y}$*)* is the loss function, N is the total number of samples, *y_i_* is the vector representing the ground truth, and ${\hat{y}}_{i}$ represents the model’s predictions.


**Stage 2: Decision Tree training with deep features**


A decision tree is a model used to solve classification and regression problems in machine learning and statistics. This model represents data in a tree structure, where each node represents a feature, while branches and leaves represent decisions and results.

In the decision tree classifier, the root node is created first. The root node represents all of the training data. A splitting criterion is then chosen to divide the data into smaller subsets. This criterion is used to increase data purity or to optimize a specific objective. In the current study, the Gini index was used and is expressed as follows (Equation (3)):(3)$$Gini\left(D\right)=1-\sum_{i=1}^{c}{\left(p_{i}\right)}^{2}$$
where *Gini(D)* is the Gini index of the node, *c* is the number of classes, and *p_i_* is the probability for the given node *i*. A node with a lower Gini index is considered purer and can be used as a better splitting criterion. It then classifies the data by creating branches and leaves of the tree. This process is performed iteratively in a recursive manner, splitting the dataset into smaller subsets and calculating new splitting criteria for those subsets. During the construction of the decision tree, the depth of the tree, the splitting criteria, and other hyperparameters are checked. At the end of the tree construction, each leaf node is associated with a class label and this classifier classifies the new data according to these leaf nodes. The results of this model are easy to interpret and explain, so it is preferred in many applications.


**Stage 3: Test application**


In the last stage of the proposed MS-LNet model, the performance analysis of the created model was performed. During the testing phase, all parameters of the MS-LNet model are frozen, and then predictions of the samples are obtained using the MS-LNet model for samples that the model has never seen before. Finally, performance scores are obtained by comparing the obtained predictions with the real values.

## 5. Experimental Results

In the current study, we proposed a multi-task learning network based on a combination of a 1D deep convolutional neural network and decision tree classifier for SDN-based DDoS detection. All experimental work was performed on the TensorFlow-Keras library in a Python environment and on a computer with Intel i9, 64GB RAM, and Nvidia RTX 2080 Ti hardware.

As a result of the above experimental studies, a multi-task learning network (MT-LNet) based on a combination of 1D-CNN and the decision tree classifier is proposed for SDN-based DDoS detection. In the proposed MT-LNet, features are extracted from the fully connected layer of the 1D-CNN model and fed into the input of the decision tree classifier. In this way, a more effective model is presented. In the third experimental study, an MT-LNet performance analysis was performed to determine the most effective features in the proposed model. The 1D-CNN model includes a flatten layer and two fully connected layers, as presented in Section 2. Unlike the fully connected layers, the flatten layer directly includes all the features in the outputs of the convolution layers. The decision tree model is trained and tested by taking individual features from each layer to select the layer that can obtain the best features. The performance results of the third experimental study are shown in Table 2.

Table 2 presents a comparison of accuracy and other scores obtained from different layers used in experimental studies. It is worth noting that the flatten layer appears to be the most effective, producing the highest accuracy and other scores. This is due to the fact that the flatten layer directly incorporates all the features present in the convolution-layer output, ensuring that no relevant information is lost. In contrast, it appears that the FC1 and FC2 layers reformatted the features based on the softmax classifier, resulting in lower scores. However, it is worth noting that the FC2 layer provided a score that was in between the two models. This could be due to the FC2 layer partially preserving the features present in the convolution-layer output. Furthermore, Table 2 indicates that the DT classifier is stronger than the softmax classifier for DDoS detection, highlighting the importance of selecting the appropriate classifier. To gain a better understanding of the performance of each layer, the experimental studies produced confusion matrices and ROC diagrams, displayed in Figure 7 and Figure 8. These diagrams offer a more detailed visualization of the strengths and weaknesses of each layer and emphasize the significance of selecting the appropriate layer for DDoS detection.

The confusion matrix in Figure 7a presents the performance of the flatten model, which showed an impressive overall accuracy of approximately 97.80%. This indicates that the model correctly classified 97.80% of all samples, thus demonstrating its high accuracy. Upon breaking down the performance by class, the flatten model achieved a remarkable accuracy of 97.65% for the NORMAL class, accurately predicting 124 out of 127 samples. However, the model did misclassify 3 samples as UDP, which is an area for improvement. On the other hand, the flatten model performed perfectly for both the TCP and ICMP classes, achieving 100% accuracy. This is a significant achievement, as these classes are crucial for network communication, and any misclassification could lead to errors in the system. For the UDP class, the flatten model correctly predicted 92.37% of the instances (109 out of 118), but it misclassified 9 samples as NORMAL. This highlights the importance of improving the model’s performance in distinguishing between the UDP and NORMAL classes.

Moving on to the results in Figure 7b, the FC1 model demonstrated an overall accuracy of approximately 96.15%, indicating that it correctly classified 96.15% of all samples. On a class basis, the model achieved accuracy scores of 91.53% and 92.91% for the UDP and NORMAL classes, respectively. Notably, the FC1 model excelled in the TCP class with a high accuracy of 98.67%. This is an impressive accomplishment, as the TCP class is vital for reliable and error-free communication between devices. Furthermore, the FC2 model in Figure 7c achieved an overall accuracy of approximately 96.51%, aligning closely with the FC1 model. Similar to the FC1 model, it displayed strong accuracy in the TCP class, indicating that both models are effective in classifying data related to TCP communications. However, there is still room for improvement in the accuracy of the FC2 model for the UDP and NORMAL classes. The confusion matrix results demonstrate the effectiveness of the flatten, FC1, and FC2 models in classifying network traffic data. However, there is still a need for improvement in accurately distinguishing between the UDP and NORMAL classes. The models’ high accuracy in classifying TCP data is a significant achievement that can contribute to the development of more reliable and efficient communication systems.

The accuracy score of 99.33% achieved by the flatten model is a testament to its exceptional performance in accurately classifying instances across multiple classes. When we examine the scores presented in Figure 7, it becomes clear that the flatten model has outperformed the other models on a class basis and in achieving the overall accuracy value. This success can be attributed to the model’s ability to effectively process and analyze data, leading to more accurate predictions. It is noteworthy that the flatten model’s superiority is not limited to a specific class but extends across multiple classes. This is a significant advantage, as it demonstrates the model’s versatility and its ability to handle various data types with ease. This versatility is particularly useful when dealing with large datasets with numerous classes, as the flatten model can accurately classify instances with high precision. Furthermore, the ROC diagram in Figure 8 provides similar results, further emphasizing the effectiveness of the flatten model. The ROC diagram shows that the model has a high true positive rate and a low false positive rate, indicating that it can accurately identify positive instances and minimize false positives. This is crucial, as false positives can lead to erroneous conclusions and misinterpretations of data. The flatten model’s exceptional performance in accurately classifying instances across multiple classes is a testament to its effectiveness in data analysis. Its versatility, precision, and ability to handle large datasets make it a valuable tool for data scientists and researchers alike.

### Discussion

In this study, we introduced the MT-LNet hybrid model for detecting DDoS attacks on SDN-based SCADA systems. While previous research has explored DDoS-attack detection in SCADA systems, they have mostly relied on datasets from traditional SCADA systems. In contrast, we utilize a dataset obtained from an SDN-based SCADA system simulation, which better represents the current state of SCADA systems. Additionally, we propose a hybrid learning model that combines deep and machine learning models for a more accurate and robust detection of DDoS attacks. This approach distinguishes itself from previous studies that mainly rely on classical machine learning-based classification methods or deep learning approaches. By combining the strengths of both models, our MT-LNet hybrid model achieves superior performance in detecting DDoS attacks on SDN-based SCADA systems. Table 3 provides a comprehensive comparison of our proposed model and other studies in the literature, highlighting the advantages and contributions of our approach. Overall, our study presents a novel and effective solution for enhancing the security of SDN-based SCADA systems against DDoS attacks.

## 6. Conclusions

This study introduces a cutting-edge approach to detecting DDoS attacks in SDN-based SCADA systems, which utilizes a multi-stage learning framework that combines a convolutional neural network and decision tree. The aim of this research is to enhance the security of industrial control systems and make SDN-based SCADA systems more resilient to DDoS attacks. The experimental results demonstrate the effectiveness of the proposed approach, as it successfully identifies and classifies attacks with high accuracy rates and low false positives, giving organizations the advantage of securing the operation of their industrial control systems. However, while this approach shows promise, more data collection and analysis are needed to understand how it performs against complex and advanced DDoS attacks. The scalability and performance of the proposed approach also require further investigation to determine its real-world applicability. Overall, this study is a significant step towards securing industrial control systems and defending industrial infrastructures against cyber threats.

In order to fully comprehend the significance of the study results, it is essential to consider them within the broader context of industrial control system security. This means taking into account the complex network of systems and processes that are involved in the operations of industries such as manufacturing, energy, and transportation. The security of these systems is critical, as any breach or disruption could have severe consequences for both human safety and the economy as a whole. It is, therefore, crucial to explore, in depth, the implications of these study findings and how they can be applied to enhance the security of industrial control systems. By doing so, we can ensure that these vital systems are protected against potential threats and that industries can operate safely and efficiently.

## Figures and Tables

**Figure 1 sensors-24-01040-f001:**
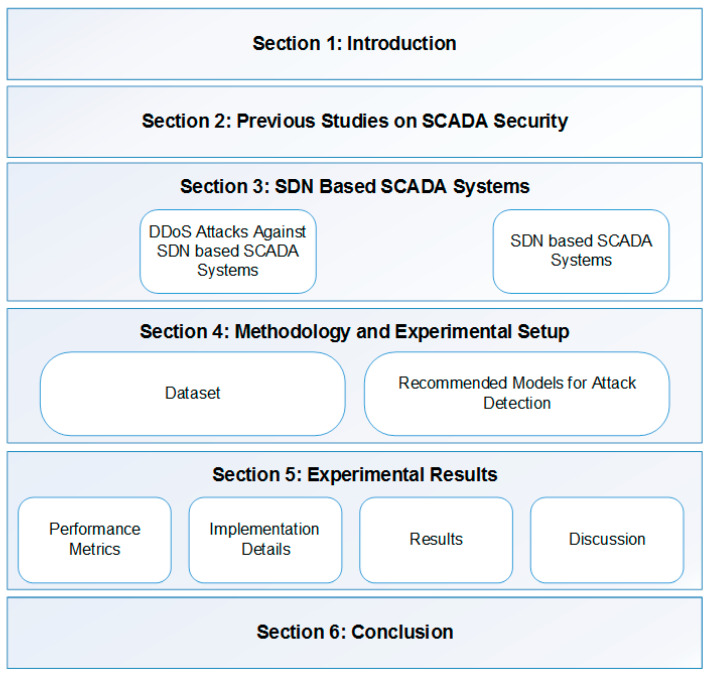
Organization of this study.

**Figure 2 sensors-24-01040-f002:**
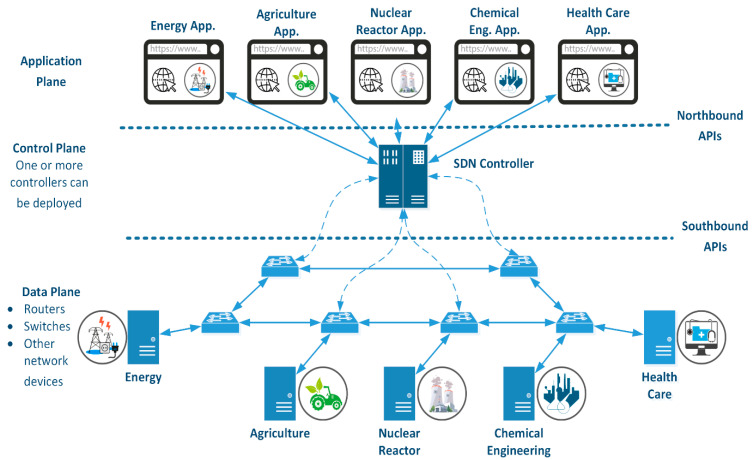
SDN-based SCADA system architecture.

**Figure 3 sensors-24-01040-f003:**
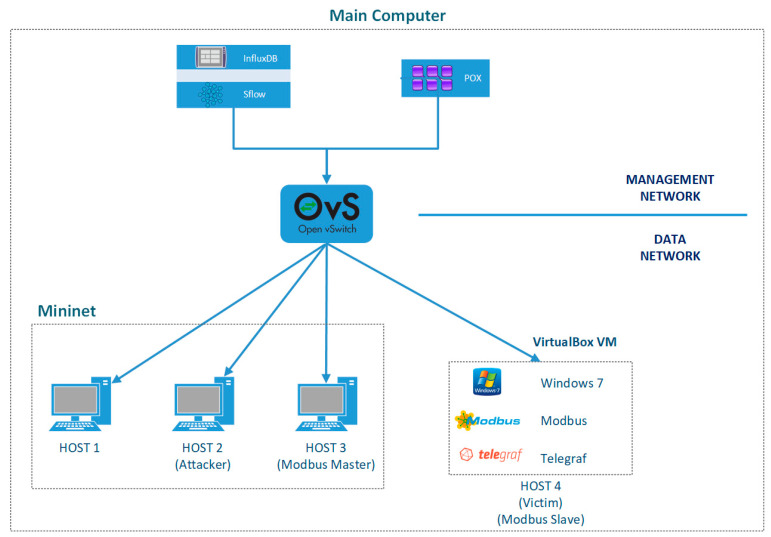
Experimental topology for network data collection of SDN-based SCADA system.

**Figure 4 sensors-24-01040-f004:**
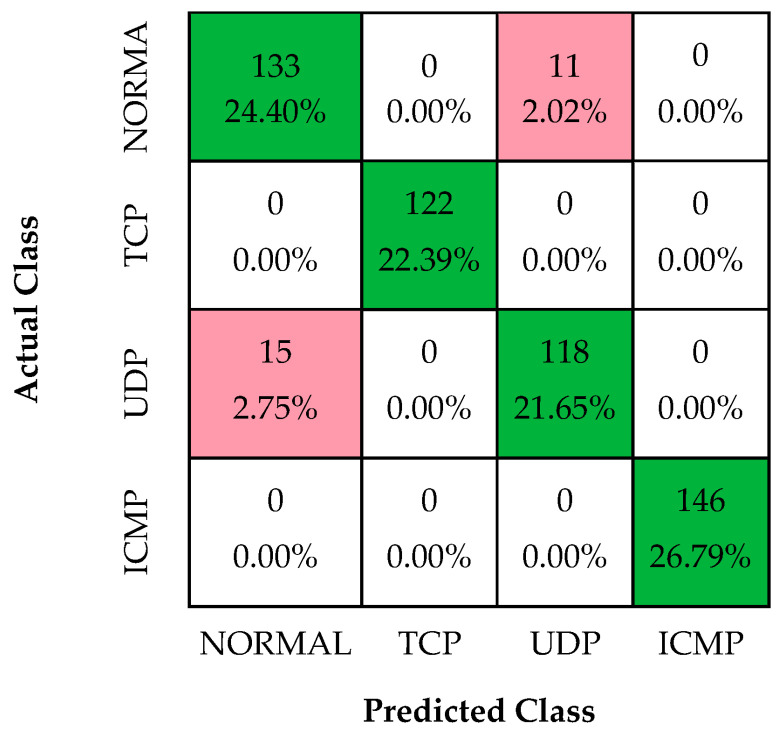
Confusion matrices of DT classifier.

**Figure 5 sensors-24-01040-f005:**
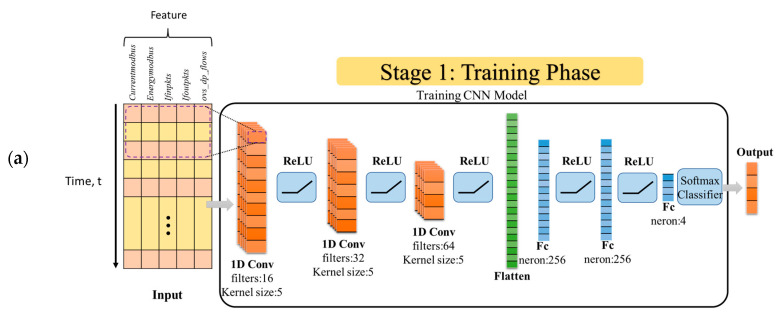
MS-LNet model-implementation steps: (**a**) Training phase of the CNN model, (**b**) Training phase of DT classifier, (**c**) Testing phase of the proposed model.

**Figure 6 sensors-24-01040-f006:**
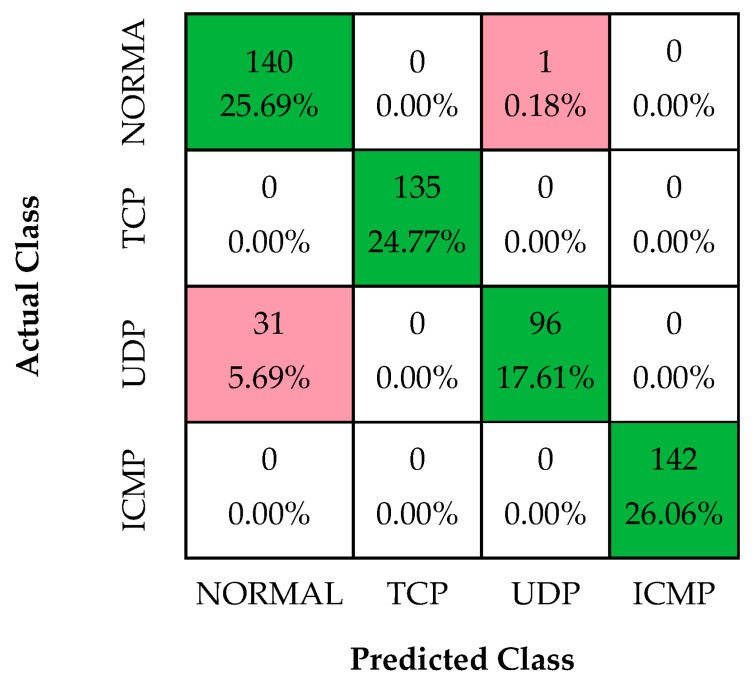
Confusion matrices of 1D-CNN.

**Figure 7 sensors-24-01040-f007:**
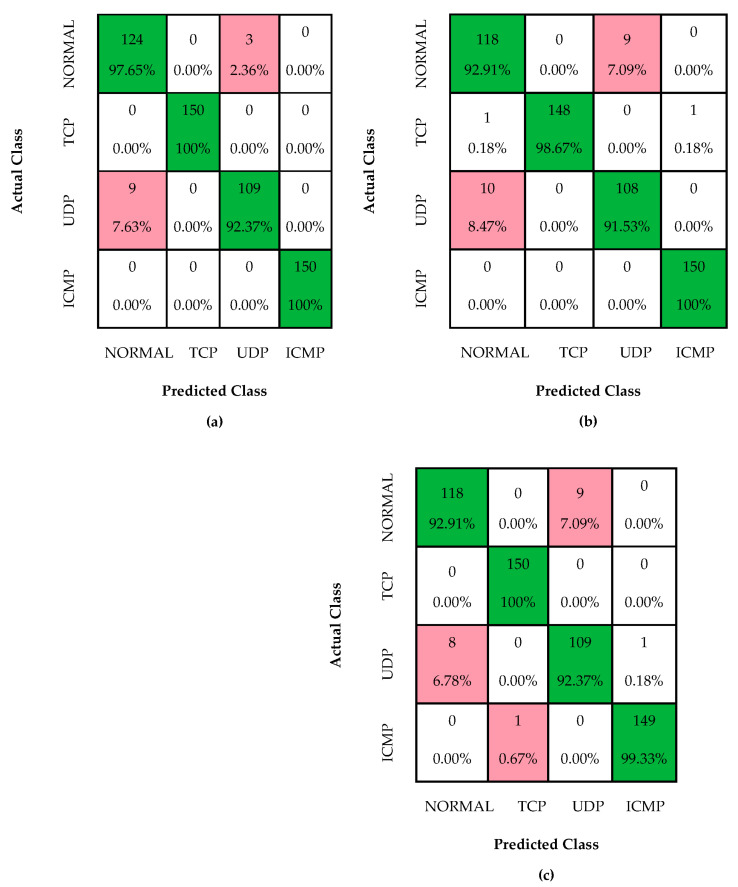
Confusion matrices of (**a**) Flatten, (**b**) FC1, and (**c**) FC2.

**Figure 8 sensors-24-01040-f008:**
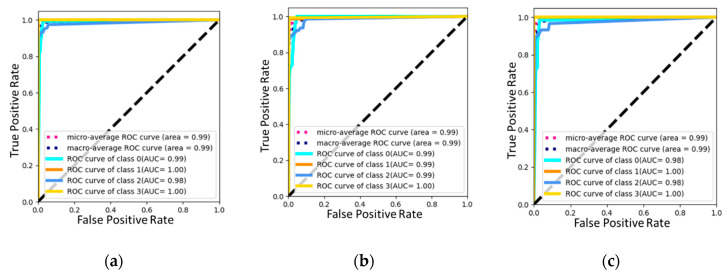
ROC diagram of (**a**) flatten, (**b**) FC1, and (**c**) FC2.

**Table 1 sensors-24-01040-t001:** Proposed 1D-CNN network architecture: fc, fully connected layer; Conv1D, 1D convolution.

Layer	Params	Output Size
Input	Input Model	89×1
Conv1D	filters: 16, kernals: 5	85×16
ReLU		86×16
Conv1D	filters: 32, kernals: 5	81×32
ReLU		82×32
Conv1D	filters: 64, kernals: 5	77×64
ReLU		78×64
Flatten		4928
Dense	nerons: 256	256
ReLU		256
Dense	nerons: 256	256
ReLU		256
Dense	nerons: 4	4
Softmax	Output Model	4

**Table 2 sensors-24-01040-t002:** Performance results (%) of the proposed model.

Model	Accuracy	Recall	Precision	F1_Score	Specificity
Flatten	97.80	97.80	97.84	97.79	99.35
FC1	96.15	96.15	96.17	96.15	98.86
FC2	96.51	96.51	96.50	96.51	98.96

**Table 3 sensors-24-01040-t003:** Comparison of previous studies in the literature.

Ref.	Datasets	ML Algorithms	Accuracy (%)
[12]	NSL-KDD and CICIDS2017	KD-TCNN	Average 98
[13]	CICIDS-2017	GRU+CNN	99.7
[14]	Mississippi State University SCADA Laboratory	NetStack	Average 93
[15]	CIC-DDoS2019 and TON_IoT	CNN RNN DNN	Average 98
[14]	Mississippi State University SCADA Laboratory	NetStack	97.36
[15]	Their own dataset	XGBoost	99
[16]	ICS	Deep autoencoder	-
[28]	Their own dataset	Autoencoder	95
[20]	Their own dataset	PRU and DT	98.5
Proposed Study	Proposed dataset in this paper	1D-CNN and decision tree-based learning model	97.80

## Data Availability

Data are contained within the article.

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
