# Peer review of "Multi-Stage Learning Framework Using Convolutional Neural Network and Decision Tree-Based Classification for Detection of DDoS Pandemic Attacks in SDN-Based SCADA Systems"

_sensors, 2024, doi:10.3390/s24031040_

Round 1
Reviewer 1 Report
Comments and Suggestions for Authors
REVIEW OF
Multi-Stage Learning Framework Using Convolutional Neural Network and Decision Tree Based Classification for Detection of DDoS Pandemic Attacks in SDN-based SCADA Systems
BY
Onur POLAT, Muammer, Hüseyin POLAT, Saadin OYUCU, Hüseyin ÜZEN, Fahri YARDIMCI and Ahmet AKSOZ
The article offers a solution to one very specific problem of information security. The task is to detect a DDoS attack in a computer network built on SDN technology and using a SCADA control system. The authors, quite possibly justifiably, believe that SDN and SCADA make a computer network specific enough to require a special protection system. The solution is to analyze traffic using two typical algorithms – convolutional neural network and decision tree-based classification. For training and performance analysis, a mock-up is created and an attack simulator is used.
The format of the presented research is traditional for such tasks, the result presented by the authors is quite consistent with other solutions in this field. In general, there is no doubt about the effectiveness of the proposed method.
The article leaves an ambivalent impression. On the one hand, there is obviously a result, this result is consistent with other studies in this field, the solution is well presented, there is a lot of high-quality graphic material. On the other hand, the article does not agree well with traditional ideas about scientific publication. So:
1. The large introduction is devoted to presenting general facts about computer networks, SDN, SCADA, DDoS attacks and the troubles they cause. The text looks more like an essay in a popular science publication. Agree, it is unlikely that the article will be read by readers who do not know what SDN and DDoS are.
2. Figure 1. For the first time the reviewer meets with such a colorful table of contents of a scientific article.
3. A good section 2. The reviewer assumes that this section is the Introduction.
4. Figure 2. What scientific information does this colorful image carry?
5. Figure 3. What scientific information does this colorful image carry?
6. Figure 4. What scientific information does this colorful image carry?
7. In connection with Figure 5 and the results of the study, questions arise: how much does the topology of the computer network used affect the result? is it necessary for a network of a different topology to train the network again? will a network trained on one topology work on another topology?
8. Section 4 has generally understandable content, but it looks too verbose. For example, who of the potential readers needs formulas (1) and (2)? However, the remaining formulas (3)-(5) are the most typical expressions for neural network technologies and it is difficult to imagine a reader of this article who will find something unknown in these ratios. In addition, it is unclear why the authors in connection with these formulas do not indicate the primary sources, where are the references?
9. Finally, the biggest scientific question is the main result – Table 4. This question should be addressed to the authors of this article, but also to the authors of the cited studies. What conclusion should the reader draw from this table? The reviewer sees one obvious conclusion: each of the researchers selected the data set that showed the effectiveness of his method in his model. How do I compare the results with each other?
The reviewer hopes that the authors will radically revise their text, significantly shorten it, getting rid of populism and turning the text into a neat scientific work. It is also expected that the results of additional experiments will be presented, which will provide answers, at least illustrative, to questions 7 and 9.
Author Response
The authors would like to express their gratitude to the reviewer for their invaluable comments and suggestions. We have carefully considered all your valuable and constructive comments that have improved the quality and presentation of our paper. Additionally, we would like to provide our explanations in response to the reviewer's comments.

Reviewer 2 Report
Comments and Suggestions for Authors
This paper proposes a framework for DoS attack in Scada systems with CNN. The overall structure of the paper is interesting and has a value for a possible publication. I have following minor issues to be considered:
1- It might be great to introduce more recent related papers specifically from MDPI journals, such as (https://www.mdpi.com/2076-3417/13/19/11067)
2- The authors have a similar paper on same journal (https://www.mdpi.com/1424-8220/24/1/155), please specify the differences, advantages of the current study over the recent previous one.
3-There is a strong need that the results presented in Fig. 9 and 10 must be detailed. Why you got these results?
4- It also may be beneficial to provide a picture of the CNN model implemented with all layers used.
5- The number of citations in introduction is poor, so please improve it.
Comments on the Quality of English LanguageMinor issues
Author Response

(The authors gave the same response as above.)

Round 2
Reviewer 1 Report
Comments and Suggestions for Authors
The authors answered the reviewer's questions, but the article as a whole retains a popular science character. There are no gross errors and inaccuracies in the text, so the reviewer leaves the final decision on the expediency of publication to the editorial board.
Author Response
We are grateful to reviewers for their valuable recommendations and corrections. We believe that the manuscript has been improved significantly. I have tried to respond all comments as much as possible. If any point missing, could you please inform me for further improvement? We will be very glad if our research manuscript is published Sensors. Thank you for all.
Best regards.
